# TRPV1 Activation by Capsaicin Mediates Glucose Oxidation and ATP Production Independent of Insulin Signalling in Mouse Skeletal Muscle Cells

**DOI:** 10.3390/cells10061560

**Published:** 2021-06-21

**Authors:** Parisa Vahidi Ferdowsi, Kiran D. K. Ahuja, Jeffrey M. Beckett, Stephen Myers

**Affiliations:** School of Health Sciences, College of Health and Medicine, University of Tasmania, Newnham Drive, Launceston, TAS 7248, Australia; parisa.vahidiferdowsi@utas.edu.au (P.V.F.); kiran.ahuja@utas.edu.au (K.D.K.A.); jeffrey.beckett@utas.edu.au (J.M.B.)

**Keywords:** capsaicin, TRPV1, CAMKK2, ERK1/2, glucose oxidation, ATP, skeletal muscle

## Abstract

Background: Insulin resistance (IR), a key characteristic of type 2 diabetes (T2DM), is manifested by decreased insulin-stimulated glucose transport in target tissues. Emerging research has highlighted transient receptor potential cation channel subfamily V member (TRPV1) activation by capsaicin as a potential therapeutic target for these conditions. However, there are limited data on the effects of capsaicin on cell signalling molecules involved in glucose uptake. Methods: C2C12 cells were cultured and differentiated to acquire the myotube phenotype. The activation status of signalling molecules involved in glucose metabolism, including 5’ adenosine monophosphate-activated protein kinase (AMPK), calcium/calmodulin-dependent protein kinase 2 (CAMKK2), extracellular signal-regulated protein kinases 1 and 2 (ERK1/2), protein kinase B (AKT), and src homology phosphatase 2 (SHP2), was examined. Finally, activation of CAMKK2 and AMPK, and glucose oxidation and ATP levels were measured in capsaicin-treated cells in the presence or absence of TRPV1 antagonist (SB-452533). Results: Capsaicin activated cell signalling molecules including CAMKK2 and AMPK leading to increased glucose oxidation and ATP generation independent of insulin in the differentiated C2C12 cells. Pharmacological inhibition of TRPV1 diminished the activation of CAMKK2 and AMPK as well as glucose oxidation and ATP production. Moreover, we observed an inhibitory effect of capsaicin in the phosphorylation of ERK1/2 in the mouse myotubes. Conclusion: Our data show that capsaicin-mediated stimulation of TRPV1 in differentiated C2C12 cells leads to activation of CAMKK2 and AMPK, and increased glucose oxidation which is concomitant with an elevation in intracellular ATP level. Further studies of the effect of TRPV1 channel activation by capsaicin on glucose metabolism could provide novel therapeutic utility for the management of IR and T2DM.

## 1. Introduction

T2DM is a major metabolic disorder that has an enormous global impact on human health [1,2,3]. This debilitating disease is characterised by increased IR, a systemic disorder of compromised ability of insulin to regulate glucose homeostasis in target peripheral tissues, including liver, adipose tissue and skeletal muscle [4].

Uncontrolled diabetes and chronic hyperglycaemia are associated with complications that reduce life expectancy [5,6]. The risk of diabetes-related complications can be averted by strategies that prevent or reduce the progression of IR before pancreatic beta-cell failure. In this context, current anti-diabetic medications are associated with various side effects which reflect substantial limitations in the current pharmacotherapy of T2DM [7,8,9]. Therefore, it is critical to understand novel options and their molecular pathways to enhance the development of innovative therapies that have robust efficacy, with fewer or no side effects for the treatment and management of T2DM.

Emerging research on the influence of other nutrients on the regulation of glucose metabolism has highlighted capsaicin, a phenolic active component of chilli peppers, as a potentially effective therapy in IR and T2DM [10]. Capsaicin activates the transient receptor potential cation channel subfamily V member 1 (TRPV1) channel [11]. TRPV1 is a nonselective calcium-permeable cation channel, involved in thermogenesis and pain-sensing and is expressed in high metabolic rate tissues [12,13,14,15]. TRPV1 has diverse biological roles that make this channel a potential therapeutical target in various pathophysiological conditions, including IR and T2DM [16].

TRPV1 activation leads to increased intracellular calcium levels and several cellular responses [15]. Elevated intracellular calcium level stimulates intracellular signalling pathways by CAMKK2 [17]. CAMKK2 activation in response to increased cytosolic calcium activates phosphorylates AMPK, an energy-sensing regulatory kinase in glucose and fat metabolism [18]. AMPK activation triggers a switch from anabolic pathways (fatty acid synthesis, sterol synthesis, protein synthesis, and glycogen synthesis) to ATP-producing catabolic pathways (fatty acid oxidation, mitochondrial biogenesis, glycolysis, and glucose uptake) [19]. Moreover, AMPK activation by 5-amino-4-imidazole carboxamide riboside (AICAR) enhances glucose uptake independently of insulin and causes an increase in insulin sensitivity in the muscle [20,21]. AMPK activation also downregulates the phosphorylation and activation of ERK1/2 which improves glucose uptake in skeletal muscle cells [22,23]. 

The ERK1/2-mediated cascade is a central signalling pathway that regulates various cellular processes including cell proliferation, differentiation, and apoptosis. ERK1/2 activity is elevated in adipose tissue of rodents and humans in IR and diabetes states [24] and is involved in the incorporation of diabetes-induced cardiac pathological changes [25]. Moreover, ERK1/2 is shown to act as a positive regulator of endoplasmic reticulum (ER) stress in IR, and administration of the ERK1/2 inhibitor, U0126, suppresses ER stress-induced IR in human embryonic kidney (HEK293) cells [22]. This suggests that inhibition of the ERK1/2 pathway could be a potential novel therapeutic strategy for IR and T2DM. 

In vitro and in vivo studies suggest beneficial effects of TRPV1 activation by capsaicin on glucose homeostasis through several mechanisms [10,26]. These include the enhancement of energy metabolism and insulin sensitivity through the reduction of inflammatory factors and elevation of fatty acid oxidation, stimulation of insulin secretion from pancreatic beta-cells, and the activation of cell signalling molecules that are essential for glucose uptake [10,17,18,19,22,27]. Regular consumption of chilli-containing foods has been shown to reduce postprandial hyperinsulinemia in humans [28]. Capsaicin regulates cellular lipid content by increasing fatty acid oxidation through the TRPV1-induced calcium influx and CAMKK2/AMPK pathway in HepG2 cells [29,30]. Moreover, capsaicin increases plasma insulin levels and reduces fasting glucose and inflammatory factors in animal models [10,31]. It is also shown that capsaicin administration stimulates glucose uptake in mouse C2C12 cells in an insulin-independent manner by activation of AMPK and its downstream kinase, p38 mitogen-activated protein kinase (p38 MAPK), in these cells [32]. Insulin-independent pathways are critical in the control of IR and provide an alternative mechanism whereby the defect in insulin-dependent signalling could be bypassed and result in an improvement in glycaemic control independent of insulin in skeletal muscle cells [33]. For example, activation of AMPK mediates an increase in glucose uptake independent of insulin which could promote glucose uptake in insulin-resistant muscle cells [34].

While attempts have been made to identify how capsaicin acts through the TRPV1 channel to improve glucose uptake, the exact molecular mechanisms remain unclear [29,30,32]. Therefore, further research is required on the mechanism of action for capsaicin-activated TRPV1 and its effects on glucose uptake.

In the present study, we showed that capsaicin causes an increase in glucose oxidation independent of insulin signalling molecules. To the best of our knowledge, this study demonstrated for the first time capsaicin-induced CAMKK2 activation and ERK1/2 phosphorylation reduction in mouse myotubes. Moreover, we also showed that pharmacological blocking of TRPV1 using SB-452533 reduces phosphorylation of CAMKK2 and AMPK that was concomitant with a reduction in glucose oxidation and ATP levels. In addition, capsaicin treatment diminished phosphorylation of ERK1/2 in mouse myotubes.

## 2. Materials and Methods

### 2.1. Reagents and Antibodies

Capsaicin, AICAR, and SB-452533 (TRPV1 antagonist) were purchased from Sigma Aldrich, Australia. Antibodies pAMPKα (Thr172) (cat#2535), AMPKα (cat#5831), pCaMKK2 (Ser511) (cat#12818), CAMKK2 (cat#16810), p44/42 MAPK (pErk1/2) (Thr202/Tyr204) (cat# 9101), p44/42 MAPK (Erk1/2) (cat#9102), pAKT (Ser473) (cat#4060), pAKT (Thr308) (cat#13038), AKT (cat#4685), pSHP-2 (Tyr580) (cat#5431), SHP2 (cat#3397), GAPDH (cat#5174), p-GSK-3β (Ser9) (cat#9322), GSK-3β (cat#9315), and HRP-linked secondary antibodies (Anti-rabbit (cat#7074) and Anti-mouse (cat#7076)) were purchased from Cell Signaling Technology, Beverly, MA, USA. TRPV1 antibody (cat#PA1-29421) and the ATP assay kit were obtained from Thermo Fisher, Melbourne, Australia.

### 2.2. Differentiation Markers and TRPV1 mRNA Expression

A total of 2 × 10^5^ cells/well were seeded in 12-well plates and differentiated to myotubes for up to 96 h. Total RNA was extracted using ISOLATE II RNA Mini Kit (Bioline, Sydney, Australia) as per manufacturer’s instructions, and cDNA was synthesised from 1 μg of total RNA using High-Capacity cDNA Reverse Transcription Kit (Thermo Fisher, Melbourne, Australia). Quantitative real-time PCR was performed using the SensiFAST SYBR^®^ No-ROX Kit (Bioline, Sydney, Australia) and the StepOnePlus™ Real-Time PCR System (Thermo Fisher, Applied Biosystems, Melbourne, Australia). PCR primers are as follows: Myogenin (*Myog*), 5′-CCTTAAAGCAGAGAGCATCC-3 and 5′- GGAATTCGAGGCATAATATGA; contractile protein troponin 1 (*Tnni1)*, 5′- GCCTATGCGCACACCTTTG-3′ and 5′-CGGGTACCATAAGCCCACACT; contractile protein troponin 2 (*Tnni2)*, 5′- AAATGTTCGAGTCTGAGTCCTAACTG-3′ and 5′- GCCAAGTACTCCCAGACTGGAT-3′; metabolic genes ATP-binding cassette subfamily A member 1 (*Abca1)*, 5′- GCTCTCAGGTGGGATGCAG-3′ and 5′- GGCTCGTCCAGAATGACAAC-3′; and fatty acid-binding protein *(**Fabp3)*, 5′- CCCCTCAGCTCAGCACCAT-3′ and 5′-CAGAAAAATCCCAACCCAAGAAT-3′ [35] *Eef2*, 5′- CACAATCAAATCCACCGCCAT-3′ and 5′- TGGCCTGGAGAGTCGATCA-3′ *Trpv1*, 5′-CCGGCTTTTTGGGAAGGGT-3 and 5′- GAGACAGGTAGGTCCATCCAC-3′ [36]. All PCR samples were assayed in triplicate and normalised to the housekeeping gene EEF2. The average cycle threshold (Ct) value of the three replicates was used for the comparative Ct (ΔΔCt) method, and the fold-change values were performed using the 2^−ΔΔCt^ method. [37].

### 2.3. Cell Culture

Skeletal muscle (C2C12) cells (Sigma Aldrich, Melbourne, Australia) were cultured in 10% fetal calf serum, and 1% Penicillin-Streptomycin-supplemented DMEM containing 4500 mg/L glucose, L-glutamine, sodium pyruvate, and sodium bicarbonate (Sigma Aldrich, Melbourne, Australia) and grown in an optimal condition of 5% CO_2_ at 37 °C. Once the cells reached 70% confluency, the C2C12 skeletal muscle myotubule phenotype was obtained by mitogen withdrawal (2% horse serum, Thermo Fisher, Melbourne, Australia) for 72 h. Three hours before the treatment, the C2C12 cells were exposed to a serum-free medium. Cells were then treated with various combinations of capsaicin, insulin, AICAR, and SB-452533 as per the following procedures.

### 2.4. Cell Viability Assay

A total of 5 × 10^3^ cells/well were seeded into 96-well plates and differentiated to myotubes for 72 h. Following the differentiation, the cells were treated with 0, 10, 20, 50, 100, 200, and 400 μM of capsaicin for up to 120 min before the cytotoxicity effect of capsaicin was assessed using the Vybrant^®^ MTT Cell Proliferation Assay Kit (Thermo Fisher, Melbourne, Australia) according to the manufacturer’s instructions. Briefly, the cell media was replaced with a media containing 10% MTT solution and incubated at 37 °C for 4 h. Following the labeling of cells with MTT, 50 μL of DMSO (Sigma Aldrich, Melbourne, Australia) was added to each well and incubated for 10 min at 37 °C. The absorbance was measured at 540 nm using a TECAN infinite M200 PRO plate reader (Tecan, Melbourne, Australia) and the cell viability of each group was calculated with comparison to the control group (0 μM of capsaicin).

### 2.5. Immunoblot Analysis of Protein Expression

A total of 4 × 10^5^ cells/well were seeded in 6-well plates and differentiated to myotubes for 72 h. The cells were then treated with 100 and 200 μM of capsaicin, 10 nM insulin, 1 and 2 mM AICAR, and different concentrations of SB-452533 for up to 60 min. The cells were lysed with RIPA buffer (Thermo Fisher, Melbourne, Australia) containing phosphatase and protease inhibitor cocktail, as per the manufacturer’s instructions. Total cellular protein was extracted by centrifuging lysates at 15,000 rpm for 10 min at 4 °C and total protein concentration was measured using a Pierce^TM^ BCA Protein Assay Kit as per the manufacturer’s instructions (Thermo Fisher, Melbourne, Australia). A total of 20 µg of protein was loaded onto 4–15% Mini-PROTEAN^®^ TGX™ Precast Protein Gels (Bio-Rad, Sydney, Australia) and transferred to a PVDF membrane using a semi-dry Turbo Transfer System (Bio-Rad, Sydney, Australia). The membranes were blocked in 5% skim milk in TBST buffer (20 mM Tris, 150 mM NaCl, 0.1% Tween20, pH 7.5) for 120 min followed by overnight incubation with the specific primary antibody (pAMPKα (1:1000 dilution), AMPKα (1:2000 dilution), pCaMKK2 (1:1000 dilution), CAMKK2 (1:2000 dilution), pERK1/2 (1:1000 dilution), ERK1/2 (1:2000 dilution), pAKT (1:1000 dilution), AKT (1:2000 dilution), pSHP-2 (1:1000 dilution), SHP-2 (1:2000 dilution), GAPDH (1:3000 dilution), TRPV1 (1:1000 dilution), p-GSK-3β (1:1000 dilution), GSK-3β (1:2000 dilution)) at 4 °C. The membranes were washed with TBST three times (5 min each time) and were then incubated with HRP-conjugated anti-rabbit/mouse secondary antibodies (1:5000 dilution) for 120 min. The membranes were washed again with TBST as previously described, and the protein bands were visualised using SuperSignal™ West Femto Maximum Sensitivity Substrate (Thermo Fisher, Melbourne, Australia) and a Fuji LAS-3000 imaging system (Berthold, Australia). Non-phosphorylated proteins were used to normalise phosphorylated protein expression except for TRPV1, which was normalised with GAPDH. Densitometric analysis of western blots was performed using ImageJ software.

### 2.6. Pharmacological Inhibition of TRPV1 and Immunoblot Analysis of AMPK and CAMKK2 Phosphorylation

Cells were grown in 6-well plates and differentiated to myotubes for 72 h. Subsequently, 0, 50, 100, 200, and 400 μM of SB-452533 (TRPV1 antagonist) was added to cells and incubated for 30 min. Cells then were treated with 200 μM capsaicin for an additional 30 min, and Western blots for p-AMPKα, AMPKα, p-CAMKK2, and CAMKK2 were performed as previously described.

### 2.7. Glucose Oxidation Assay

Cells were seeded at 5 × 10^3^ cells/well in 96-well plates and differentiated to myotubes for 72 h. Twenty-four hours before the assay, the myotubes medium was replaced with Minimum Essential Medium Eagle glucose-free media (Sigma Aldrich, Melbourne, Australia). On the day of the assay, myotubes were treated for 30 min with 200 μM SB-452533100 followed by 60 min treatment with 100 and 200 µM capsaicin, and 10 nM insulin (positive control for glucose oxidation). The glucose oxidation assay was performed using the Glucose Uptake-Glo^TM^ Assay Kit (Promega, Sydney, Australia). Briefly, 50 μL of 0.1 mM 2DG was added to cells and incubated for 30 min at 25 °C. Then 25 μL of stop buffer and neutralization buffer were added to each well and the plate was shaken briefly. Finally, 100 μL of 2DG6P was added to each well, and the plate was incubated for 60 min at 25 °C. Then, luminescence was recorded with 0.3–1 s integration on the TECAN infinite M200 PRO plate reader and glucose oxidation level was calculated.

### 2.8. Intracellular ATP Measurement

Cells were cultured into 96-well plates and differentiated to myotubes for 72 h. Cells were treated with 0, 100, and 200 μM of capsaicin for up to 120 min. Myotubes also were treated with 200 μM SB-452533 for 30 min followed by another 30 min incubation with 200 μM of capsaicin to evaluate intracellular ATP content changes due to capsaicin and SB-452533 treatment. ATP was quantified by bioluminescence assay using an ATP determination kit (ThermoFisher, Melbourne, Australia), following the manufacturer’s instructions. 

### 2.9. Statistical Analysis

Data were analysed by the *t*-test using GraphPad Prism 8 and expressed as the mean ± Standard Deviation (SD). *p* < 0.05 represents a statistically significant difference between two groups.

## 3. Results

### 3.1. The Trpv1 mRNA and Protein Are Expressed in Skeletal Muscle Cells during the Differentiation

Proliferating myoblasts can be differentiated into post-mitotic multinucleated myotubes by mitogen withdrawal. The muscle-specific contractile and metabolic phenotype was assessed by gene expression of several important markers including the differentiation-specific hierarchical basic helix loop regulator *Myog*, *Tnni1*, *Tnni2*, *Abca1*, and *Fabp3* [34]. *Myog*, *Tnni1*, and *Tnni2* mRNA expressions following 48 h of mitogen withdrawal were significantly increased over those of time 0 (start point of differentiation) (Figure 1a–c). Similarly, the metabolic genes *Abca1* and *Fabp3* were significantly induced following 96 h of mitogen withdrawal (Figure 1d,e). These data confirmed that the C2C12 skeletal muscle cells had acquired the differentiated and metabolic phenotype. Based on these results and literature, we used 72 h for C2C12 differentiation [32].

We next tested for the expression of *Trpv1* in mRNA and protein level in C2C12 cells during differentiation. The expression for *T**rpv1* mRNA and TRVP1 protein were similar during the differentiation of the C2C12 skeletal muscle cells (Figure 2a–c).

### 3.2. Capsaicin Treatment Did Not Affect Mouse Myotubes Viability

To determine if capsaicin treatment affected cell viability, mouse C2C12 cells were differentiated to myotubes and treated with varying concentrations of capsaicin (0/control group, 10, 20, 50, 100, 200, and 400 μM) for up to 120 min and an MTT assay was performed as previously described on three independent experiments. The percentage of the viable cells was then calculated. It was observed that capsaicin treatment did not significantly affect cell viability at all concentrations over 60 min (Figure 3). However, there was a significant reduction in cell viability in the 400 µM capsaicin-treated cells at 120 min. Accordingly, the 100 and 200 μM concentrations of capsaicin were selected for the subsequent experiments. These concentrations are also further supported in the literature [32].

### 3.3. Capsaicin Increases the Phosphorylation of Calmodulin (CAMKK2) and AMPK but Diminishes the Phosphorylation of ERK1/2

CAMKK2 mediates AMPK phosphorylation and subsequently increases the expression and translocation of glucose transporter type 4 (GLUT4) to the cell membrane which leads to glucose disposal into the cells through an insulin-independent pathway [18,38]. Conversely, ERK1/2 phosphorylation is reported to downregulate the AMPK pathway and impair glucose uptake in skeletal muscle cells [22,39]. To assess CAMKK2, AMPK, and ERK1/2 phosphorylation status by capsaicin in skeletal muscle cells, we treated cells with 100 and 200 μM of capsaicin over 60 min. We observed a significant increase in phosphorylation of AMPK over 60 min in 100 and 200 µM capsaicin-treated cells (Figure 4a,b). Capsaicin also significantly increased phosphorylation of CAMKK2 at 15, 30, and 60 min of treatment (100 µM and 200 µM capsaicin) (Figure 4c,d), while 100 and 200 µM capsaicin diminished the phosphorylation of ERK1/2 over 60 min incubation. We also tested the phosphorylation status of AMPK in the presence of AICAR, the well-known AMPK activator [38]. As shown in Figure 4g, 1 and 2 mM AICAR led to phosphorylation of AMPK after 30 min of treatment.

### 3.4. Capsaicin Does Not Activate Insulin Signalling Molecules

To test whether capsaicin has an insulin-mimetic effect in mouse myotubes, we tested the ability of capsaicin to activate three important molecules involved in the insulin signalling pathway. AKT is a serine/threonine-protein kinase that is activated by insulin to stimulate various cellular processes including glucose metabolism [40]. SHP-2 plays a critical role as a signal regulator in insulin-responsive cells through binding to the tyrosine-phosphorylated insulin receptor substrate proteins [41]. Accordingly, we treated myotubes with 100 and 200 μM of capsaicin over 60 min. Western blot analysis demonstrated that capsaicin does not phosphorylate and activate AKT and SHP-2 over 60 min of treatment in mouse skeletal muscle cells (Figure 5a,b). As mentioned before, insulin promotes glucose uptake in skeletal muscle cells by the phosphorylation of AKT, a key molecule in the insulin signalling pathway [42]. To verify our cell system robustness, we treated cells with 10 nM insulin. Figure 5c demonstrates that 10 nM of insulin activated AKT after 30 min of treatment.

### 3.5. Capsaicin Increases Glucose Oxidation in Mouse Myotubes

It has been previously reported that capsaicin-stimulated AMPK activation leads to an increase in glucose uptake in mouse skeletal muscle cells [32]. We also aimed to confirm that capsaicin’s action causes an enhancement of glucose disposal in the mouse myotubes. Accordingly, we measured glucose oxidation in the myotubes treated with 100 and 200 μM capsaicin over 60 min. As demonstrated in Figure 6a, 100 and 200 μM capsaicin significantly elevated glucose oxidation in myotubes over 60 min.

### 3.6. Capsaicin Does Not Affect Phosphorylation of GSK-3β But Increases ATP Production in Mouse Myotubes

Following glucose uptake in skeletal muscle cells, glucose molecules can be added to the glycogen chain for storage through glycogenesis or produce ATP during glycolysis [43]. Glycogen synthase kinase-3 β (GSK-3β) is a serine/threonine kinase that is involved in several biological processes including glucose and glycogen metabolism and insulin action in skeletal muscle cells [44]. Insulin inhibits the activation of GSK-3β through the PI3K/AKT pathway which subsequently promotes glycogen synthase (GS) activity and glycogen synthesis in skeletal muscle cells. To evaluate the effect of capsaicin in GSK-3β regulation in mouse skeletal muscle cells, we treated cells with 100 and 200 μM capsaicin and 10 nM insulin over 60 min. Then we measured the expression of p-GSK-3β and GSK-3β in the presence or absence of capsaicin and insulin. The Western blot analysis demonstrated that capsaicin treatments do not significantly affect GSK-3β phosphorylation. However, insulin deactivated GSK-3β during the three different time points (Figure 6c,d). As earlier described, AMPK activation enhances energy metabolism and glycolysis which leads to ATP synthesis in the cells [10]. To determine the effect of capsaicin-induced ATP generation in mouse myotubes, we treated cells with 100 and 200 μM capsaicin for 5, 15, 30, 60, and 120 min and intracellular ATP level was measured. As illustrated in Figure 6b, there is a reduction in ATP synthesis after 15 min incubation with 100 μM of capsaicin followed by an elevation in ATP levels over 60 min treatment. Capsaicin (200 μM) did not affect the ATP level after 15 min incubation but caused an increase in ATP generation between 30 and 120 min of incubation (Figure 6b).

### 3.7. Capsaicin’s Action in Phosphorylation of AMPK and CAMKK2, and Elevation of Glucose Oxidation and ATP Production Is through the Activation of TRPV1 Channel in Mouse Myotubes

As described previously, TRPV1 activation stimulates various signalling molecules including AMPK and CAMKK2 that consequently regulate many cellular events [12]. To test whether AMPK and CAMKK2 phosphorylation by capsaicin is mediated through TRPV1 activation, we treated cells with different concentrations of the TRPV1 inhibitor, SB-452533, for 30 min followed by 200 μM capsaicin treatment for another 30 min. Our data demonstrate that capsaicin-induced phosphorylation of AMPK and CAMKK2 is significantly reduced by increasing concentrations of SB-452533 treatment (Figure 7a–d). We also demonstrated that 200 μM SB-452533 inhibits ATP production after 30 min and reduces glucose oxidation after 60 min incubation in mouse myotubes (Figure 7e,f).

## 4. Discussion

In the present study, we demonstrated that 100 and 200 µM capsaicin administration leads to the phosphorylation of CAMKK2 and AMPK and reduces ERK1/2 phosphorylation. This was concomitant with an increase in glucose oxidation and ATP production in mouse C2C12 skeletal muscle cells, which occurred in an insulin-independent manner and through the activation of the TRPV1 channel (Figure 8).

Although the concentrations of capsaicin in this study are higher than blood capsaicin concentrations (~0.2 µM) previously reported, it is important to understand the mechanisms of capsaicin action on glycaemic control that may lead to the identification of novel therapeutic targets for insulin resistance and type 2 diabetes using capsaicin analogues or adjunct therapies that bypass defective insulin signalling pathways [45]. Capsaicin reduces blood glucose level by enhancing insulin levels in diabetic rats [46]. Similarly, analogues of capsaicin are efficacious in lowering blood glucose and improving glycaemic control. For example, capsiate (a capsaicin analogue) is shown to reduce body weight gain, visceral fat accumulation, and improve glucose tolerance in diabetic rats [47]. Another capsaicin analogue (resiniferatoxin) also appears to prevent the deterioration of glucose homeostasis in early diabetic rats [48]. Furthermore, overtly diabetic rats administered with resiniferatoxin markedly improved glucose tolerance and had an increased insulin response to glucose [48]. While the above studies were performed in rodents, there are beneficial effects of chilli consumption in humans in the context of improving glucose homeostasis. For instance, consumption of foods containing chilli showed a beneficial effect on glucose homeostasis in healthy adults by improving postprandial hyperglycaemia and hyperinsulinemia [28]. More specifically, chilli supplementation also improved postprandial hyperglycaemia and hyperinsulinemia in women with gestational diabetes mellitus. Although there were some reported minor side effects including mild diarrhea, a heat sensation in the oral cavity, and skin weals, these reverted to normal after 2 or 3 days of supplementation [49].

Capsaicin improves glucose homeostasis by increasing insulin sensitivity and through the activation of important signalling molecules and the reduction of inflammatory factors [10,32]. Capsaicin has previously been shown to increase glucose uptake independent of insulin signalling molecules such as insulin receptor substrate 1 (IRS-1) and AKT phosphorylation in C2C12 cells [10]. Similarly, our results confirmed that capsaicin’s action in glucose metabolism is independent of activation of insulin signalling molecules such as AKT and SHP-2 in the differentiated C2C12 cells.

AMPK activation promotes glucose uptake, glycolysis, and ATP production in skeletal muscle cells [50]. Our data showed that capsaicin treatment activates AMPK and leads to an increase in glucose oxidation and ATP production in the mouse myotubes. This is in line with previous in vivo and in vitro reports in different tissues. A study on obese mouse models demonstrated that capsaicin administration promotes mitochondrial function and increased ATP levels in these animals [51]. Moreover, capsaicin appears to also increase glucose metabolism and ATP production in the human intestinal epithelium [10,52]. We showed that capsaicin-induced glucose oxidation and ATP production occur via the TRPV1 channel activation. Pharmacological inhibition of the TRPV1 channel led to a reduction in pAMPK that was concomitant with a decrease in glucose oxidation and ATP production in capsaicin-treated mouse myotubes. Previous studies have reported that TRPV1 activation has beneficial effects on energy and glucose homeostasis [16]. TRPV1 activation by capsaicin reduces blood glucose concentration in mice during exercise [53] while activated TRPV1 also increases glucose uptake in mice on high-fat diets [54]. In the present study, we demonstrated that capsaicin promotes glucose metabolism by increasing glucose oxidation and intracellular ATP production through the activation of TRPV1 channel in mouse myotubes. These results support the beneficial role of the TRPV1 channel in glucose metabolism and suggest the potential therapeutic value of TRPV1 in glucose metabolism disorders such as IR and T2DM.

A previous study reported that capsaicin activates AMPK and its downstream kinase, p38 MAPK, leading to glucose uptake in C2C12 cells, and hypothesised that reactive oxygen species (ROS) is the possible upstream activator of AMPK; however, the upstream signalling pathway of AMPK phosphorylation remains elusive [32]. Liver kinase B1 (LKB1) and CAMKK2 are two well-established upstream kinases implicated in the phosphorylation of AMPK on Thr172 [55]. CAMKK2 is activated in response to intracellular calcium elevation and is involved in glucose uptake in skeletal muscle cells [17,56]. It is reported that capsaicin-mediated AMPK activation occurs through calcium and CAMKK2 activation in HepG2 cells [30]. To the best of our knowledge, this is the first study to show that 100 and 200 μM of capsaicin increases CAMKK2 phosphorylation in mouse myotubes. We also demonstrated that pharmacologically inhibiting TRPV1 blocks CAMKK2 activation. Together, these results suggest that capsaicin-mediated activation of TRPV1 and subsequent phosphorylation of CAMKK2 and AMPK are involved in the regulation of glucose metabolism in mouse myotubes. 

ERK1/2 activation impairs glucose uptake in smooth muscle cells by blocking IRS-1 and subsequently insulin signalling pathway in these cells [57,58]. ERK1/2 activity is also reported to be elevated in adipose tissue of diabetic humans and rodents [24]. Moreover, inhibitory cross-talk between AMPK and ERK1/2 pathways is associated with ER stress and ER stress-induced IR in skeletal muscle cells [22]. Inhibition of ERK1/2 signalling by chemical inhibitors reverses ER stress, elevates phosphorylation of AMPK, increases glucose uptake, and eventually improves IR in myotubes [22]. This suggests that inhibition of ERK could be a potential novel therapy for IR and T2DM. For the first time, our data demonstrated that 100 and 200 μM of capsaicin inhibits ERK1/2 phosphorylation. This effect of capsaicin on the downregulation of ERK1/2 could improve glucose homeostasis through the improvement of AMPK activity and inhibition of ER stress in myotubes [22,23].

GSK3β is an important signalling molecule involved in a variety of cellular processes including glucose transport and metabolism. It is a key regulator of GS activity. The inhibition of GS by GSK3β causes a decrease in glycogen synthesis and a concomitant increase in blood glucose. Insulin inhibits GSK3β activity by triggering phosphorylation of an inhibitory serine of this kinase, in liver and skeletal muscle cells [59] and this inhibition is associated with an increase in glucose uptake in skeletal muscle [60]. There is an important potential role for GSK3β in the development of IR and T2DM [61,62]. GSK3β is overexpressed in type 2 diabetic humans and is associated with impaired glucose disposal in these patients [63]. Moreover, GSK3β inhibitors have useful therapeutic effects on IR and T2DM [62,63]. We evaluated the effect of capsaicin on GSK3β phosphorylation in mouse myotubes and observed that capsaicin does not significantly affect GSK3β phosphorylation. This is in agreeance with previous work which reported that GSK3β phosphorylation was not affected by capsaicin administration in rat skeletal muscles [44].

TRPV1 is expressed in metabolically active tissues including liver, adipose, and skeletal muscle [10]. Capsaicin-mediated activation of TRPV1 that facilitates cellular responses resulting in glycaemic homeostasis, strengthens TRPV1 as an attractive target and capsaicin as a beneficial molecule for the study of mechanisms that improve metabolic disorders including insulin resistance and type 2 diabetes [10]. Although several studies have been conducted to determine the molecular pathways involved in glucose metabolism that are activated by capsaicin through the TRPV1 channel, numerous molecules and pathways remained unknown. The presented study for the first time demonstrated that capsaicin activates CAMKK2 in mouse myotubes which is concomitant with inhibition in ERK1/2 phosphorylation. This study also showed for the first time that pharmacological inhibition of TRPV1 blocks AMPK and CAMKK2 phosphorylation, glucose oxidation, and ATP production in mouse skeletal muscle cells. Future studies on capsaicin-mediated activation of TRPV1 could lead to the development of novel and more effective therapies for the management and treatment of IR and T2DM.

## 5. Conclusions

To conclude, in this study, we demonstrated that TRPV1 activation by capsaicin causes an increase in glucose oxidation and ATP generation in mouse skeletal muscle cells. We also showed that capsaicin activates key regulators of glucose metabolism, CAMKK2, and AMPK in mouse myotubes. Moreover, for the first time, we demonstrated that capsaicin has an inhibitory effect on ERK1/2 phosphorylation which could promote glucose uptake in skeletal muscle cells by improving AMPK action in these cells. In contrast to AMPK and CAMKK2, insulin signalling molecules, AKT, and SHP-2 were not activated by capsaicin. Finally, for the first time, we showed that pharmacological inhibition of TRPV1 using its antagonist, SB-452533, inhibits capsaicin’s action in the phosphorylation of AMPK and CAMKK2, ATP generation, and increase in glucose oxidation in the mouse myotubes.

## Figures and Tables

**Figure 1 cells-10-01560-f001:**
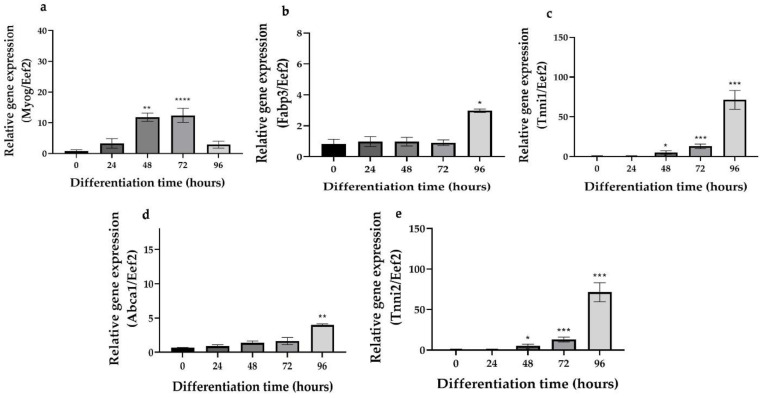
Gene expression of *Myog, Tnni1*, *Tnni2*, *Abca1,* and *Fabp3*. (**a**–**e**). Quantitative real-time PCR mRNA expression of *Myog*, *Tnni1*, *Tnni2*, *Abca1*, and *Fabp3* respectively during the differentiation of C2C12 skeletal muscle cells. Data are presented as the mean ± SD of three independent repeats (*n* = 3). * *p* < 0.05; ** *p* < 0.01; *** *p* < 0.001; **** *p* < 0.0001 indicate a significant difference between differentiated and the control groups.

**Figure 2 cells-10-01560-f002:**
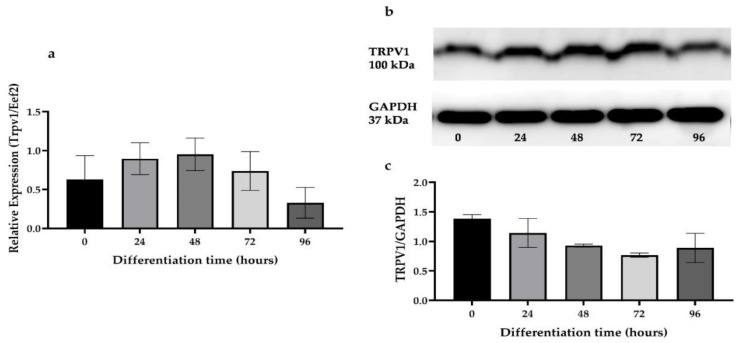
Expression of *Trpv1* mRNA and protein levels during skeletal muscle cell differentiation. (**a**) Quantitative real-time PCR mRNA expression of *Trpv1* during C2C12 skeletal muscle cell differentiation. (**b**,**c**) Western blot results and densitometric analysis of TRPV1 protein during the C2C12 differentiation, respectively. Data are presented as the mean ± SD of three independent repeats (*n* = 3).

**Figure 3 cells-10-01560-f003:**
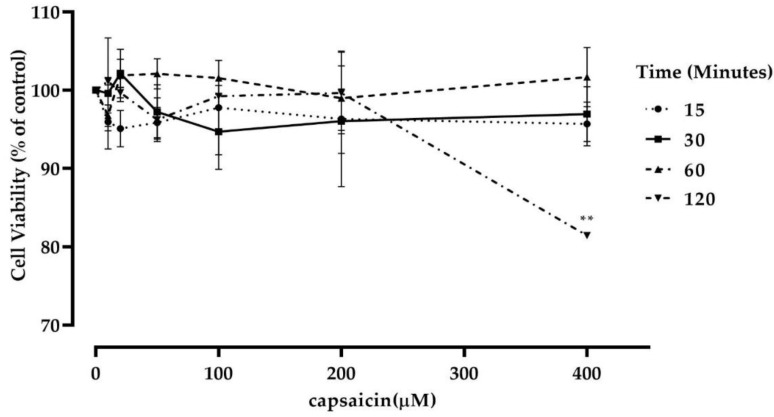
Effect of capsaicin treatment on C2C12 skeletal muscle cell viability. Data are presented as the mean ± SD of three independent experiments (*n* = 3). ** *p* < 0.01 indicates a significant difference between capsaicin-treated and control groups.

**Figure 4 cells-10-01560-f004:**
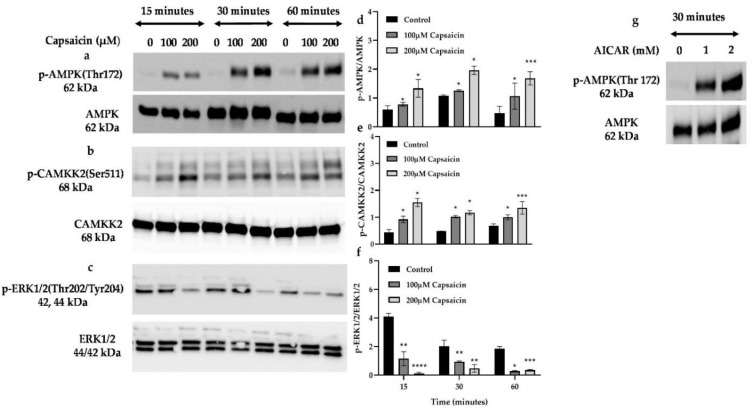
Western blot analysis of phosphorylated and total AMPK, CAMKK2, and ERK 1/2 in capsaicin treated mouse myotubes. (**a**,**b**,**d**,**e**) Densitometric analysis for the pAMPK/AMPK and pCAMKK2/CAMKK2 over 60 min. (**c,f**) Representative Western blot and densitometric analysis for the ERK1/2 in the absence or presence of capsaicin over 60 min. (**g**) Immunoblots of p-AMPK and AMPK protein expression in the AICAR treated myotubes after 30 min incubation. Data are presented as mean ± SD of three independent repeats (*n* = 3). * *p* < 0.05, ** *p* < 0.01, *** *p* < 0.001, and **** *p* < 0.0001 indicate a significant difference between capsaicin-treated and control groups.

**Figure 5 cells-10-01560-f005:**
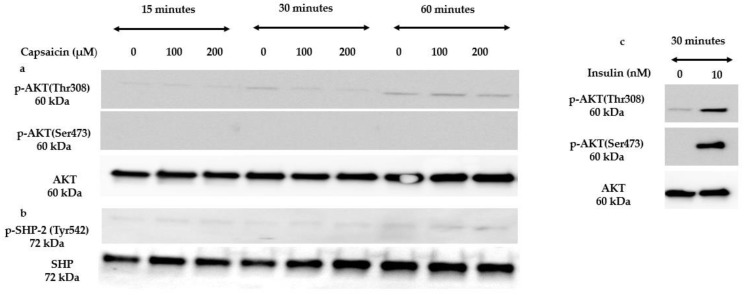
Western blot results of insulin signalling molecules in the presence of capsaicin over 60 min. (**a**,**b**) Representative Western blots for the AKT and SHP-2 phosphorylation in mouse differentiated C2C12 cells treated with 100 and 200 µM capsaicin over 60 min. (**c**) Immunoblots of p-AKT and AKT protein expression in the 10 nM insulin-treated myotubes after 30 min.

**Figure 6 cells-10-01560-f006:**
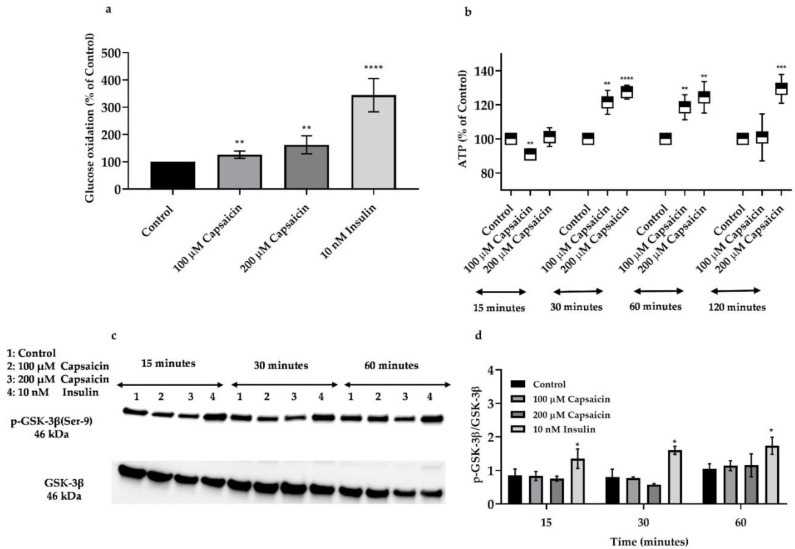
Glucose oxidation levels, ATP levels, and Western blot analysis for phosphorylated and total GSK-3β in capsaicin and insulin-treated mouse skeletal muscle cells. (**a**) Glucose oxidation levels following 60 min incubation with 100 and 200 μM of capsaicin and 10 nM insulin. (**b**) Intracellular ATP generation in the 0, 100, and 200 μM capsaicin treated cells over 120 min incubation. (**c**) Representative Western blot for p-GSK-3β in the presence and absence of capsaicin and insulin over 60 min incubation. (**d**) Relative densitometric bar graph of GSK-3β in the capsaicin and insulin-treated mouse skeletal muscle cells over 60 min. Data are presented as mean ± SD of three independent repeats (*n* = 3). * *p* < 0.05, ** *p* < 0.01, *** *p* < 0.001, and **** *p* < 0.0001 indicate a significant difference between capsaicin/insulin-treated and control groups.

**Figure 7 cells-10-01560-f007:**
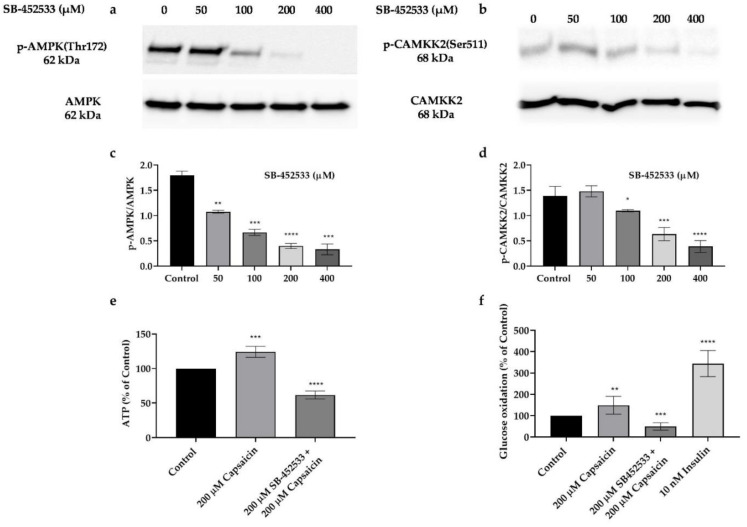
Western blot analysis for phosphorylated and total AMPK and CAMKK2, glucose oxidation and intracellular ATP levels in SB-452533 and capsaicin treated mouse skeletal muscle cells**.** (**a**–**d**) Western blot and densitometric analysis of p-AMPK, AMPK, p-CAMKK2, and CAMKK2 with increasing concentrations of SB-452533 in the presence of 200 μM capsaicin. (**e**) Intracellular ATP level in the 0 and 200 μM capsaicin treated myotubes in the presence or absence of 200 μM SB-452533. (**f**) Glucose oxidation percentage in 200 μM capsaicin treated myotubes in the absence or presence of 200 μM SB-452533 and 10 nM insulin after 60 min incubation time. Data are presented as mean ± SD of three independent repeats (*n* = 3). * *p* < 0.05, ** *p* < 0.01, *** *p* < 0.001, and **** *p* < 0.0001 indicate a significant difference between control and SB-452533-treated groups.

**Figure 8 cells-10-01560-f008:**
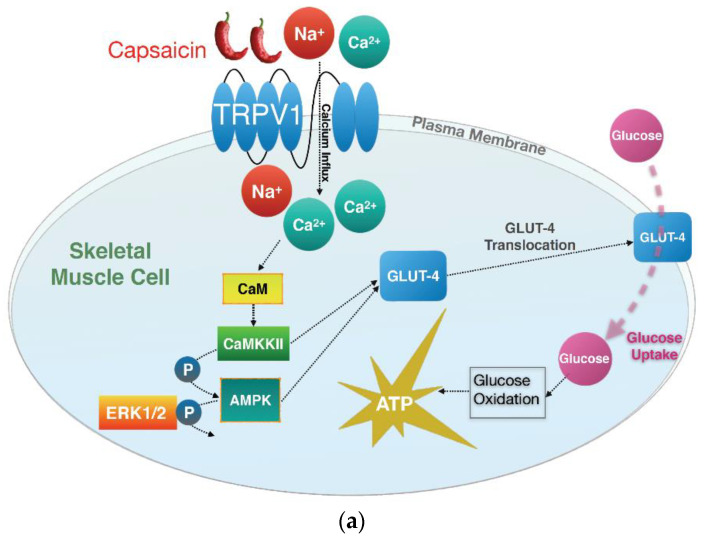
Graphical summary of capsaicin-induced signal transduction in mouse skeletal muscle cells. (**a**) Activation of CAMKK2 and AMPK, dephosphorylation of ERK1/2, glucose oxidation, and ATP production by capsaicin in skeletal muscle cells. (**b**) Pharmacological blocking of TRPV1 and subsequent inhibition of downstream signalling events, glucose oxidation, and ATP production in skeletal muscle cells.

## Data Availability

The data presented in this study are available in the manuscript.

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
