# Peer review of "TRPV1 Activation by Capsaicin Mediates Glucose Oxidation and ATP Production Independent of Insulin Signalling in Mouse Skeletal Muscle Cells"

_cells, 2021, doi:10.3390/cells10061560_

Round 1
Reviewer 1 Report
General comments:
The revised manuscript still does not provide enough new knowledge. The authors may wish to consider the following specific points to enhance the thrust of their work.
Specific comments:
1) Is glucose uptake affected by capsaicin in the myotubes? If so, how?
2) Which glucose catabolism pathways are induced in the C2Cl12-derived myotubes?
3) Through which bioenergetic pathways was ATP generated in capsaicin-treated C2C12 cells? Substrate-level ATP production or oxidative phosphorylation, or both?
4) Figure 7f. Capsaicin-mediated glucose oxidation is small compared to insulin-mediated glucose oxidation, and since capsaicin does not stimulate insulin signaling (Figure 5), capsaicin will not act to restore insulin sensitivity. This questions the extent with which capsaicin could benefit individuals with T2DM.
5) Discuss the physiological relevance of 100-200 uM capsaicin in humans.
6) Further studies are needed to tease out the contribution of AMPK in the mechanism of capsaicin-initiated glucose oxidation. How indispensable is AMPK to the mechanism of action of capsaicin on glucose oxidation and ATP production? Additional loss-of-function studies are required to address this question.
7) Once AMPK is activated, it should conserve ATP by downregulating ATP-consuming biosynthetic pathways. Which of these pathways were downregulated in C2C12 cells, and with what functional consequences for the myotubes?
8) What is responsible for ERK1/2 inhibition in the first place? Which of ERK1/2 inhibition, CAMKK2 activation or AMPK activation occurs first? This question will require time course studies.
9) A summary of capsaicin signaling in the form of a graphical figure would be useful at the end of the manuscript.
10) Figure 6 caption: spelling of “insulin”.
Author Response
Reviewer 1 comments:
General comments:
The revised manuscript still does not provide enough new knowledge. The authors may wish to consider the following specific points to enhance the thrust of their work.
Authors Response
Thank you for your valuable comments. In the revision, we have added additional information and a figure (Figure 8) as you have suggested to improve the quality of the manuscript. Below we have addressed each of your comments.
Comment 1: Is glucose uptake affected by capsaicin in the myotubes? If so, how?
Authors Response
Our study showed that 100 and 200 µM capsaicin caused an increase in glucose oxidation in myotubes after 30 minutes incubation (please see Figure 6). A previous study also showed that capsaicin elevates glucose uptake in C2C12 skeletal myotubes through the activation of AMPK (1). Although we did not test glucose uptake mechanisms as it was not the focus of our studies, we have acknowledged the previous important study by Kim et al. in our manuscript discussion (lines 344-346).
- Kim, S.H., Hwang, J.T., Park, H.S., Kwon, D.Y. and Kim, M.S., 2013. Capsaicin stimulates glucose uptake in C2C12 muscle cells via the reactive oxygen species (ROS)/AMPK/p38 MAPK pathway. Biochemical and biophysical research communications, 439(1), pp.66-70.
Comment 2: Which glucose catabolism pathways are induced in the C2Cl12-derived myotubes?
Authors Response
The major glucose catabolism pathway in all eukaryotes is glycolysis. The pentose phosphate pathway (PPP) is another glucose oxidative pathway, but its primary role is anabolic, rather than catabolic. It is responsible for synthesis of the nucleotide precursor ribose and nicotinamide adenine dinucleotide phosphate (NADPH) required for glucose metabolism (1). The PPP is mostly restricted to tissues and cells implicated in extensive fatty acid synthesis (liver, adipose, and mammary gland) or active synthesis of cholesterol and steroid hormones (liver, adrenal glands, and gonads) (2). In skeletal muscle, the activity of the PPP is low compared to other tissues, and the PPP only becomes active in skeletal muscle in response to damage to provide substrates for muscle repair (3).
- Choi, J., Kim, E-S., and Koo, JS. 2018. Expression of pentose phosphate pathway-related proteins in breast cancer. Disease Markers, Article ID 9369358; https://doi.org/10.1155/2018/9369358
- Nelson, D. L., & Cox, M. M. (2017). Lehninger principles of biochemistry (7th ed.). W.H. Freeman. Chapter 14.
- Evans, PL., McMillin, SL., Weyrauch, LA., and Witczak, CA. 2019. Regulation of skeletal muscle glucose transport and glucose metabolism by exercise training. Nutrients, 11:2432; doi:10.3390/nu11102432.
Comment 3: Through which bioenergetic pathways was ATP generated in capsaicin-treated C2C12 cells? Substrate-level ATP production or oxidative phosphorylation, or both?
Authors Response
Extensive mitochondrial-based experiments and equipment are needed to address this comment and the present study does not focus on ATP production mechanisms by capsaicin. These are studies that could be done in future manuscripts that emphasise capsaicin-induced ATP production mechanisms.
Comment 4: Figure 7f. Capsaicin-mediated glucose oxidation is small compared to insulin-mediated glucose oxidation, and since capsaicin does not stimulate insulin signaling (Figure 5), capsaicin will not act to restore insulin sensitivity. This questions the extent with which capsaicin could benefit individuals with T2DM.
Authors Response
As the reviewer correctly mentioned, capsaicin does not activate insulin signalling; however, our results demonstrated that capsaicin activates other important signalling molecules (CAMKK2 and AMPK) which improve glucose metabolism independent of insulin signalling (1). Our results (Figure 6) indicate that capsaicin significantly elevates glucose oxidation although its effect is less than insulin’s effect. These data are also consistent with other studies in C2C12 myotubes (2). As highlighted in the manuscript (line 70-73), understanding insulin-independent pathways is critical in the field of diabetes research as these pathways could provide alternative mechanisms to improve glucose metabolism when there is a defect in insulin receptor-mediated action (2,3).
- Williams, J.N. and Sankar, U., 2019. CaMKK2 signaling in metabolism and skeletal disease: a new axis with therapeutic potential. Current osteoporosis reports, 17(4), pp.169-177.
- Kim, S.H., Hwang, J.T., Park, H.S., Kwon, D.Y. and Kim, M.S., 2013. Capsaicin stimulates glucose uptake in C2C12 muscle cells via the reactive oxygen species (ROS)/AMPK/p38 MAPK pathway. Biochemical and biophysical research communications, 439(1), pp.66-70.
- Stanford, K.I. and Goodyear, L.J., 2014. Exercise and type 2 diabetes: molecular mechanisms regulating glucose uptake in skeletal muscle. Advances in physiology education, 38(4), pp.308-314.
Comment 5: Discuss the physiological relevance of 100-200 uM capsaicin in humans.
Authors Response
For the current project, our aim was to understand the mechanism of action of capsaicin, rather than determine the optimal concentrations in which capsaicin has a biological effect. Once the molecular mechanism of action for capsaicin-induced glucose uptake is clear, we would consider testing for the optimal doses. Also, these concentrations are consistent with previous research (1, 2).
- Bort, A. et al. (2019). The red pepper's spicy ingredient capsaicin activates AMPK in HepG2 cells through CaMKKβ. PLoS One, 14, e0211420-e0211420, doi:10.1371/journal.pone.0211420.
- Kim, S.H. et al. (2013). Capsaicin stimulates glucose uptake in C2C12 muscle cells via the reactive oxygen species (ROS)/AMPK/p38 MAPK pathway. Biochemical and Biophysical Research Communications, 439, 66-70, doi:10.1016/j.bbrc.2013.08.027.
Comment 6: Further studies are needed to tease out the contribution of AMPK in the mechanism of capsaicin-initiated glucose oxidation. How indispensable is AMPK to the mechanism of action of capsaicin on glucose oxidation and ATP production? Additional loss-of-function studies are required to address this question.
Authors Response
In the present study, we showed that capsaicin treatment activates AMPK through TRVP1 which is concomitant with an increase in ATP production and glucose oxidation. The focus of the manuscript was to determine the effect of capsaicin-mediated activation of TRVP1 and its subsequent downstream signalling on AMPK and CAMKK2. We acknowledge that further studies on capsaicin and AMPK are of interest and will require, at the least, extensive knock-out studies of AMPK in cells, and further experimentation with transgenic AMPK loss-of-function mouse models.
Comment 7: Once AMPK is activated, it should conserve ATP by downregulating ATP-consuming biosynthetic pathways. Which of these pathways were downregulated in C2C12 cells, and with what functional consequences for the myotubes?
Authors Response
We appreciate this comment however, AMPK is implicated in many complex cascading pathways implicated in energy metabolism (1). Similar to the above comment (comment 6), extensive knock-out studies of AMPK in cells, and further experimentation with transgenic AMPK loss-of-function mouse models are required and therefore these studies sit outside the scope of this study at this time.
- Carling, D. 2004. The AMP-activated protein kinase cascade – a unifying system for energy control. Trends in Biomedical Sciences, 29:18-24. doi:10.1016/j.tibs.2003.11.005
Comment 8: What is responsible for ERK1/2 inhibition in the first place? Which of ERK1/2 inhibition, CAMKK2 activation or AMPK activation occurs first? This question will require time course studies.
Authors Response
We acknowledge this comment, and it is not clear how ERK1/2 is inhibited in this study. To determine the mechanisms behind this, extensive time course/ inhibition studies are needed. Additionally, to answer this question comprehensively, loss-of-function studies for CAMKK2, TRVP1, and AMPK are required to determine the sequence of cellular events. These studies are certainly interesting to us but sit outside of the focus of this manuscript.
Comment 9: A summary of capsaicin signaling in the form of a graphical figure would be useful at the end of the manuscript.
Authors Response
Thank you very much for your positive comment. The suggested figure has been added to the manuscript (Figure 8).
Comment 10: Figure 6 caption: spelling of “insulin”.
Authors Response
Thank you for your comment. Spelling of “insulin” has been revised (line 284).
Reviewer 2 Report
The authors have improved their manuscript and the readability of the figures. It is nevertheless a pity not to have tested lower doses of capsaicin which would be in adequacy with the potential use of this molecule in humans.Author Response
Reviewer 2 comments:
The authors have improved their manuscript and the readability of the figures. It is nevertheless a pity not to have tested lower doses of capsaicin which would be in adequacy with the potential use of this molecule in humans.
Authors Response
Thank you very much for the positive feedback. As previously mentioned, for the current project our aim was to understand the mechanism of action of capsaicin, rather than determine the optimal concentrations of capsaicin in humans. We did test low concentrations of capsaicin in our cell system, but these had no effect on cell signalling events. The concentrations we used in study are also consistent with other studies (1,2). Once the mechanism of action for capsaicin-induced glucose uptake is clear, we would consider testing for the optimal doses.
- Bort, A. et al. (2019). The red pepper's spicy ingredient capsaicin activates AMPK in HepG2 cells through CaMKKβ. PLoS One, 14, e0211420-e0211420, doi:10.1371/journal.pone.0211420.
- Kim, S.H. et al. (2013). Capsaicin stimulates glucose uptake in C2C12 muscle cells via the reactive oxygen species (ROS)/AMPK/p38 MAPK pathway. Biochemical and Biophysical Research Communications, 439, 66-70, doi:10.1016/j.bbrc.2013.08.027.
Reviewer 3 Report
The resubmitted manuscript by Ferdowsi et al. has been thoroughly revised. The authors have carefully considered the comments and have not only rewritten some part of the manuscript, but also provide results of new experiments, which strengthen their argument that TRPV1 modulates glucose metabolism in C2C12 cells. Experiments with SB-452533 are particularly appreciated.
Minor comments:
- I would suggest that the molecular weight markers that were used on the gel are indicated next to the blots (rather than the predicted molecular weights of target proteins).
- C2C12 cells were grown under high glucose conditions (4.5 g/L). Could this affect the interpretation of the results? For instance, could this affect capacity for glucose oxidation or responsiveness/sensitivity to insulin?
- TRPV1 receptors are involved in sensing thermal stimuli, nociception as well as cough. Would this represent a problem if TRPV1 is targeted pharmacologically? A short comment on this issue would be very welcome. (It is relevant to mention also potential adverse effects.)
Author Response
Reviewer 3 comments:
General comments:
The resubmitted manuscript by Ferdowsi et al. has been thoroughly revised. The authors have carefully considered the comments and have not only rewritten some part of the manuscript, but also provide results of new experiments, which strengthen their argument that TRPV1 modulates glucose metabolism in C2C12 cells. Experiments with SB-452533 are particularly appreciated.
Authors Response
Thank you very much for the very positive feedback.
Comment 1: I would suggest that the molecular weight markers that were used on the gel are indicated next to the blots (rather than the predicted molecular weights of target proteins).
Authors Response
Thank you. We decided to only show the predicted size of the target protein as this is clearer and avoids crowding of the figure. The majority of manuscripts do not show the molecular weight marker for this reason.
Comment 2: C2C12 cells were grown under high glucose conditions (4.5 g/L). Could this affect the interpretation of the results? For instance, could this affect capacity for glucose oxidation or responsiveness/sensitivity to insulin?
Authors Response
Thank you for your comment. As you have correctly mentioned, growing cells in high glucose conditions could affect interpretation of the results for glucose oxidation and the cell’s responsiveness to insulin. For the glucose oxidation experiments, cells were passaged in low glucose as outlined in the Materials and Methods in Minimum Essential Medium Eagle glucose-free media. We have amended the text to include “glucose-free media” (lines 155-157).
Comment 3: TRPV1 receptors are involved in sensing thermal stimuli, nociception as well as cough. Would this represent a problem if TRPV1 is targeted pharmacologically? A short comment on this issue would be very welcome. (It is relevant to mention also potential adverse effects.)
Authors Response
We appreciate you comment. Most of the reported side effects of pharmacologically targeting TRPV1 is related to this receptor antagonists rather than its agonists like capsaicin (1,2). However, we revised the manuscript based on the reviewer’s comment and added information about the potential adverse effects of TRPV1 activation by capsaicin (line 322 and 323).
- Gram, D.X., Holst, J.J. and Szallasi, A., 2017. TRPV1: a potential therapeutic target in type 2 diabetes and comorbidities?. Trends in Molecular Medicine, 23(11), pp.1002-1013.
- Brito, R., Sheth, S., Mukherjea, D., Rybak, L.P. and Ramkumar, V., 2014. TRPV1: a potential drug target for treating various diseases. Cells, 3(2), pp.517-545.
Round 2
Reviewer 1 Report
The revised manuscript still does not provide enough new knowledge. Moreover, the authors did not reasonably justify the concentrations of capsaicin (100-200 µM) used in their study, which are not attainable in the blood by oral administration of capsaicin according to Choi et al. (2013) "Pharmacokinetic characteristics of capsaicin-loaded nanoemulsions fabricated with alginate and chitosan." J Agric Food Chem 61(9): 2096, who reported peak capsaicin blood concentration of ~60 µg/L plasma, i.e., ~0.2 µM, which is several orders of magnitude lower than 100-200 µM and was not acknowledged in the present manuscript.
Author Response
Thank you for your comment
Please see attached the response
Kind regards
Dr Stephen Myers

This manuscript is a resubmission of an earlier submission. The following is a list of the peer review reports and author responses from that submission.
Round 1
Reviewer 1 Report
The present study shows that TRPV1 activation by capsaicin activates CAMKK2 and AMPK, diminishes phosphorylation of ERK1/2, and elevates glucose oxidation and ATP levels in mouse C2C12 skeletal muscle cells. But this is not new. It has already been shown that (1) capsaicin acts through TRPV1, (2) TRPV1 activation induces calcium influx and stimulates intracellular calcium/calmodulin-dependent protein kinase 2 (CAMKK2), and (3) capsaicin administration stimulates glucose uptake in mouse C2C12 cells by activation of AMPK and its downstream kinase, as Ferdowsi et al. pointed out. Although the premise of the manuscript was to identify the molecular mechanisms of capsaicin in glucose metabolism, the amount of new information at this stage of the work is extremely limited, no new insight is provided, consequently, the manuscript does not fulfill its intended objective.
Reviewer 2 Report
In this study the authors have investigated the influence of Capsaicin on glucose oxidation through the TRPV1 2 channel, CAMKK2, and AMPK in mouse C2C12 skeletal muscle cells.
However, since it has already been shown that capsaicin increases glucose uptake and activates TRPV1, CAMKK2 and AMPK, this work is not very original and does not bring much more to the extent that the authors merely confirm previously published data....
I have some major criticisms about this paper.
Major criticisms:
1- The figures are really illegible and sloppy. Make an effort to: 1) enlarge the legends on the axes of the graphs, 2) homogenize the font size, the graph size and align them. Put yourself in the reader's shoes!
2- In your C2C12 cells you test insulin and AICAR on one side to show that they are functional. On the other hand you test Capsaicin.
It would be important to study whether the pathways are synergistic by testing: 1) Capsaicin with AICAR on the AMPK pathway and 2) Capsaicin with insulin on the phosphorylation of Akt and GSK3.
3- The concentrations of capsaicin tested in this study are really very high even if they correspond to doses already used. I think that if one day we want to consider a capsaicin-based treatment to improve insulin resistance, it would be important to test more physiological doses.
How much chilli peppers should a man eat in order to have a concentration of 200 μM of capsaicin in his blood?
Reviewer 3 Report
Ferdowsi et al. extend our understanding of the actions of capsaicin on AMPK and insulin signalling in cultured skeletal muscle cells. Their study indirectly supports the notion that TRPV1 might in skeletal muscle represent a pharmacological target for treatment of insulin resistance and type 2 diabetes. While the study presents an interesting set of results, there are some issues that should be addressed.
Comments/suggestions:
Conclusions/discussion:
- “We also showed that this was achieved through the CAMKK2-AMPK pathway which promoted glucose uptake.” Glucose uptake was not measured in this study. Also, while the study shows that capsaicin stimulates AMPK phosphorylation and ATP production, there are many calcium-dependent proteins in the cell and ATP production and glucose oxidation may not have been mediated via CAMKK2 and/or AMPK. Although it is plausible, I am therefore not convinced that an increase in ATP synthesis occurred due to activation of AMPK. To demonstrate that ATP-production was AMPK-dependent, AMPK should be inhibited or knocked-down.
An alternative approach would be an experiment with CAMKK2 inhibitor, which would demonstrate whether ATP production, glucose oxidation, and AMPK phosphorylation were increased by capsaicin due to activation of the CAMKK2. Such control experiment would be important also because Kim et al. demonstrated that AMPK can be activated in C2C12 cells by capsaicin via other mechanisms (BBRC 2013).
- TRPV1 antagonist was used only to show assess capsaicin-stimulated AMPK activation. However, it was not tested to show whether increases in ATP production and glucose oxidation are mediated via TRPV1.
- Insulin resistance was not assessed in this paper. C2C12 cells were not treated with insulin after being pretreated with capsaicin. I therefore think interpretation with regard to effects of capsaicin on insulin resistance should be cautious. Alternatively, a control experiment, in which C2C12 cells are treated with capsaicin and insulin could be done.
- TRPV1 receptors are involved in sensing thermal stimuli, nociception as well as cough. Would this represent a problem if TRPV1 is targeted pharmacologically?
Results:
- Figure 4 shows an important control experiment. However, I do not think it is justified to show this result as an independent figure since these results are well-known and established. Indeed, AICAR-stimulated AMPK activation and insulin-stimulated Akt activation are very well known facts. I recommend integrating these results into another figure (otherwise they should be omitted).
- Figures with blots: indicators of molecular weight should be included (position of molecular weight markers).
- Title of paragraph 3.5: can tyrosine really be described as insulin signalling molecule? Phospho-tyrosyl residues are part of signalling proteins.
- It is not clear what is shown of phospho-Tyrosine blot. Molecular weight markers are essential to appreciate the results with anti-phospho-Tyr antibody. Since, the authors did control experiments with insulin (Figure 4) it would be nice to show results of these experiments with anti-phospho-Tyr antibody to compare with results of capsaicin experiments.
- Only one phosphorylated band of ERK can be seen clearly. Which of two (ERK1 or ERK2) is it?
Abstract and introduction:
- Authors refer to “Insulin-independent molecules such as AMPK…” However, AMPK is not an insulin-independent molecule. However, it can stimulate glucose uptake in insulin-independent fashion.
- I would suggest to shorten the background and include more results in the Abstract. Also, abstract mentions glucose uptake, whereas glucose oxidation was measured in this study.
- Skeletal muscle accounts for up to 80% of glucose disposal during intravenous glucose application (e.g. as assessed by clamp), but not after a meal, when liver also takes up a significant fraction of ingested glucose.
- The role of AMPK activation during muscle contractions and exercise has been debated. It now seems unlikely that AMPK activation is necessary or a major mechanism leading to increased glucose uptake in skeletal muscles during exercise. For instance, see McConell, AJP Endocrinol Metab 2020; 318: E564–E567.
Methods:
- Glucose oxidation assay: the protocol description says that cells were differentiated into myoblasts. Probably this should be myotubes?
- What was glucose concentration of DMEM?
- Catalogue numbers and dilutions of antibodies should be provided.